# Comparative efficacy of antibiotic(s) alone or in combination of corticosteroids in adults with acute bacterial meningitis: A systematic review and network meta-analysis

Ajaree Rayanakorn[1,2]*, Hooi-Leng Ser[1,3], Priyia Pusparajah[1], Kok-Gan Chan[4,5]*, Bey Hing Goh[6,7], Tahir Mehmood Khan[8,9], Surasak Saokaew[10,11], Shaun Wen Huey Lee[8,12], Learn-Han Lee[1]*

1 Novel Bacteria and Drug Discovery Research Group (NBDD), Microbiome and Bioresource Research Strength, Jeffrey Cheah School of Medicine and Health Sciences, Monash University Malaysia, Bandar Sunway, Malaysia, 2 Faculty of Public Health, Chiang Mai University, Chiang Mai, Thailand, 3 Institute of Biomedical and Pharmaceutical Science, Guangdong University of Technology, Guangzhou, P.R. China, 4 Division of Genetics and Molecular Biology, Faculty of Science, Institute of Biological Sciences, University of Malaya, Kuala Lumpur, Malaysia, 5 International Genome Centre, Jiangsu University, Zhenjiang, China, 6 College of Pharmaceutical Sciences, Zhejiang University, Hangzhou, P.R. China, 7 Biofunctional Molecule Exploratory Research Group (BMEX), Biomedicine Research Advancement Centre (BRAC), School of Pharmacy, Monash University Malaysia, Bandar Sunway, Malaysia, 8 School of Pharmacy, Monash University Malaysia, Bandar Sunway, Malaysia, 9 The Institute of Pharmaceutical Sciences (IPS), University of Veterinary & Animal Sciences (UVAS), Outfall road, Lahore, Pakistan, 10 Center of Health Outcomes Research and Therapeutic Safety (COHORTS), School of Pharmaceutical Sciences, University of Phayao, Phayao, Thailand, 11 Unit of Excellence on Clinical Outcomes Research and IntegratioN (UNICORN), School of Pharmaceutical Sciences, University of Phayao, Phayao, Thailand, 12 Asian Centre for Evidence Synthesis in Population, Implementation and Clinical Outcomes, Health and Well-Being Cluster, Global Asia in the 21st Century (GA21) Platform, Monash University Malaysia, Selangor, Malaysia

* lee.learn.han@monash.edu (LHL); kokgan@um.edu.my (KGC); ajaree.rayanakorn@monash.edu, au. ajaree@gmail.com (AR)

**Data Availability Statement:** All relevant data are within the manuscript and its Supporting Information files.

## Abstract

### Objective

To compare relative efficacy of different antibiotic therapies either with or without the addition of corticosteroids among adult patients with acute bacterial meningitis on all-cause mortality, neurological complications and any hearing loss.

### Methods

We searched nine databases from inception to 8 February 2018 for randomized controlled trials evaluating pharmacological interventions and clinical outcomes in adult bacterial meningitis. An updated search from 9 February to 9 March 2020 was performed, and no new studies met the inclusion criteria. Study quality was assessed using the revised Cochrane Risk of Bias Tool. The Grading of Recommendations Assessment, Development and Evaluation system was used for quality of evidences evaluation. Meta-analyses were conducted to estimate the risk ratio with 95% confidence interval for both direct and indirect

**Funding:** This study is funded by External Industry Grants from Biotek Abadi Sdn Bhd (vote no. GBA-808138 and GBA-808813) awarded to L-HL and by the University of Malaya for Research Grant (FRGS grant FP022-2018A and HIR grant H-50001-00-A000027) awarded to K-GC. The specific roles of these authors are articulated in the 'author contributions' section. Biotek Abadi Sdn Bhd had no role in study design, data collection and analysis, decision to publish, or preparation of the manuscript.

**Competing interests:** The authors have read the journal's policy and have the following competing interests: L-HL received External Industry Grants from Biotek Abadi Sdn Bhd. There are no patents, products in development or marketed products associated with this research to declare. This does not alter our adherence to PLOS ONE policies on sharing data and materials.

comparisons on the primary outcomes of all-cause mortality, neurologic sequelae and any hearing loss. The study was registered in PROSPERO (CRD42018108062).

## Results

Nine RCTs were included in systematic review, involving 1,002 participants with a mean age ranging between 25.3 to 50.56 years. Six RCTs were finally included in the network-meta analysis. No significant difference between treatment was noted in meta-analysis. Network meta-analysis suggests that corticosteroids in combination with antibiotic therapy was more effective in reducing the risk of any hearing loss compared to mono antibiotic therapy (RR 0.64; 95%CI, 0.45 to 0.91, 4 RCTs, moderate certainty of evidence). Numerical lower risk of mortality and neurological complications was also shown for adjunctive corticosteroids in combination with antibiotic therapy versus mono antibiotic therapy (RR 0.65; 95% CI, 0.42 to 1.02, 6 RCTs, moderate certainty of evidence; RR 0.75; 95%CI, 0.47 to 1.18, 6 RCTs, moderate certainty of evidence). No differences were noted in the adverse events between different therapies. The overall certainty of evidence was moderate to very low for all primary outcomes examined.

## Conclusions

Results of this study suggest that corticosteroids therapy in combination with antibiotic is more effective than mono antibiotic therapy in reducing the risk of any hearing loss in adult patients with acute bacterial meningitis. More well-design RCTs to investigate relative effective treatments in acute bacterial meningitis particularly in adult population should be mandated to aid clinicians in treatment recommendations.

## Introduction

Acute bacterial meningitis (ABM) is a severe infection resulting in high mortality and neurologic complications. The case fatality rate (CFR) ranges between 20–30% with only partial recovery among most survivors [1]. Partly due to the implementation of vaccination programs, the disease incidence and mortality have been largely averted among children and neonates with 17.9% and 15.6% reduction during the past ten years whereas little progress has been made among those between 15–50 years old [2]. The fatality rate and the number of neurological abnormalities remain high among adults with ABM, particularly those with pneumococcal meningitis [3].

Initiation of an early and effective therapy is essential for the treatment success and avoidance of the disease mortality and morbidity. However, the choice of antimicrobial regimen in ABM varies depending on the patient population, the causative pathogens and clinical data mainly based on randomized trials in pediatrics [4] with conflicting findings and limited data in adult population.

The pathophysiology of ABM which causes inflammatory reactions at subarachnoid space has given the rationale of adjunctive corticosteroids administration in ABM treatment in reducing the inflammation caused by infection [5–12]. Yet, the benefits of treatment in reducing mortality and neurologic sequelae remains uncertain in adult population. A recent meta-analysis concluded that corticosteroids use was associated with a reduction of hearing loss and

neurological complications among patients with bacterial meningitis in high-income countries but not in low-income countries [13]. However, majority of studies included were in children. Most trials conducted varied greatly in terms of study population, interventions, and timing. The results of many studies were inconclusive with a relatively small sample size and limited data in adult patients with bacterial meningitis despite a significant burden. With limited head-to-head studies, application of network meta-analysis (NMA), an analytical approach that does not only include direct comparisons, but also indirect comparisons in addition to traditional meta-analysis that assess the treatment effects based on pair-wise head to head direct comparisons [14, 15] would be useful to assist in decisions.

Therefore, this systematic review and network meta-analysis (NMA) was performed to comprehensively examine the relative efficacy of corticosteroids in combination with antibiotic treatment, dual antibiotic, and mono antibiotic therapies for both direct and indirect comparison in treating adult bacterial meningitis by analyzing results from randomized controlled trials which are considered to be on the top level in the hierarchy of evidences.

## Methods

The systematic review and network meta-analysis was conducted following the protocol registered with PROSPERO, CRD42018108062. This study was reported according to the Preferred Reporting Items for Systematic Reviews and Meta-analyses (PRISMA) Extension for NMA [16].

### Participants

Our systematic review and network meta-analysis focused on adults with acute bacterial meningitis. These infections were evaluated by suggestive clinical picture of bacterial meningitis with a combination of turbid cerebrospinal fluid (CSF), elevated protein and decreased glucose level in CSF or microbiological proven bacterial culture and/or Gram's staining using blood and cerebrospinal fluid prior to the initiation of drug treatment. The detail of study participants is shown in S1 Appendix, eTable 3.2 Description of study participants in Appendix 3.

### Search strategy and study selection

We searched in nine relevant databases including CINAHL Plus, Cochrane Library, EMBASE, Global Health, Ovid Medline, PubMed, SciELO, Science Direct and Scopus using the MeSH terms "bacterial meningitis" OR "bacterial meningitis AND treatment" limited in human without time or language restriction until 8 February 2018 (See S1 Appendix, Appendix 1: The search string used). Additionally, we performed an updated search from 2018–2020 up to 9 March 2020 and identified no new studies to be included in our analyses. Articles would be included if they were randomized controlled trials (RCTs) evaluating on pharmacological treatment and clinical outcomes in adult bacterial meningitis with at least one of the primary outcomes (all-cause mortality, neurologic complications, hearing loss) documented. RCTs studies in children, aseptic, tuberculosis, viral or other type of bacterial meningitis, HIV and immunocompromised patients were excluded. Reference lists of the included articles and relevant systematic reviews were also checked and verified to identify additional potential articles to be included. Articles with unclear information or unavailable full-text, the corresponding authors would be contacted by at least two email attempts. However, there was no response received. Two reviewers (AR and H-LS) independently screened and selected articles by title and abstract. The full text eligibility was also assessed independently. Any discrepancies were discussed and resolved with consensus with other authors (PP, K-GC, BHG, T-MK, SS, SWHL, and L-HL) to reach agreement.

## Data extraction

Data extraction was performed by two independent reviewers (AR and H-LS) with further confirmation with other investigators (PP, K-GC, BHG, TMK, SS, SWHL, and L-HL). Any disagreements were resolved by a consensus among all authors. Information including the first author's name and year of publication, the mean age, patients' characteristics, the study location, causative pathogen, the number of participants and study design, treatment, and outcomes were extracted. Studies with at least one of any primary outcomes were included and all reported events would be extracted. Documentation as no event of either primary outcome would be done for studies without that event reported or with complete recovery among all participants.

## Quality assessment

The revised Cochrane Risk-of-bias Tool for RCTs (RoB 2.0) was used to assess the quality of included randomized control trials [17]. The Grading of Recommendations Assessment, Development, and Evaluation (GRADE) assessment using GRADEpro GDT software (GRADE Working Group, McMaster University, Hamilton, ON, Canada) [18] was used to assess the quality of evidence [19]. In the GRADE framework, the five domains including risk of bias, inconsistency, indirectness, imprecision, and publication bias were used to rate the quality of evidence from pairwise and network meta-analysis into four levels: high, moderate, low, and very low [20].

## Outcomes and definitions

The primary outcomes were all-cause mortality, neurologic sequelae and any hearing loss. Neurologic sequelae were defined as focal neurologic abnormalities other than hearing loss or seizure or pre-existing conditions before the disease onset as well as the Computed Tomography (CT) brain abnormality at discharge or latest follow-up identified by a complete physical examination. Hearing loss was defined as any degree of hearing impairment either unilateral or bilateral assessed by audiologic examination upon discharge or the last follow-up.

The secondary outcomes were adverse events defined as clinical evidences of undesirable outcomes after therapies initiation including gastrointestinal bleeding, arthritis, endocarditis, herpes simplex viral or fungal infections, recurrent or persistent fever at a temperature of 38 Degree Celsius or higher.

## Data synthesis and meta-analysis

The data from all included studies were summarized descriptively based on mean/median ± standard deviation for continuous variables. Dichotomous variables were summarized by proportion or event rate. Standard inverse-variance random-effect meta-analysis using STATA 14.2 (College Station, Texas, USA) within frequentist framework was done to combine all outcome data across RCTs comparing direct interventions between corticosteroids plus antibiotic therapy, dual antibiotic therapy, and mono antibiotic therapy. Heterogeneity in each pairwise was tested employing $I^2$ statistics [21, 22].

## Network meta-analysis

A network meta-analysis (NMA) was performed using the network commands in STATA to combine all direct and indirect comparisons. The three major assumptions: (1) similarity, (2) consistency and (3) transitivity were checked [23]. The geometry network maps were drawn to give an overview of the relationships between pairs of treatments (corticosteroids in

combination with antibiotic, dual antibiotic therapy, and mono antibiotic therapy) [23]. The assumptions of consistency and inconsistency models were investigated [23]. The network forest and interval plots to summarize an effect size (ES) as pooled risk ratio (RR) with a 95% confidence interval (CI) were illustrated to measure the mixed treatment effects. Contribution plots were drawn to estimate the percentage of each comparison to the entire network [24]. The surface under the cumulative ranking area (SUCRA) was calculated to identify the hierarchy of superiority among interventions [23, 24].

### Sensitivity analysis and publication bias

We initially planned to perform a sensitivity analysis to examine the potential impact of high risk of bias studies and subgroup analysis among studies conducted in high vs. low- and middle-income countries (LMICs). However, none of these approaches could be performed due to a small number of studies could be included in the analyses.

The publication bias was assessed by comparison-adjusted funnel plot and trim-and-fill method for pairwise meta-analysis.

## Results

### Study selection

Overall 10,993 records were identified through database searches and 3 records were identified through other sources (Fig 1). A total of 8,653 records were screened after removing duplicates. Of 39 potentially relevant articles reviewed in full-text, 9 studies were included in the systematic review, while 6 studies were incorporated in NMA study. The full workflow of current literature search is shown in S1 Appendix, Appendix 1, eFigure 1.1. The details of 30 studies excluded after full-text review were presented in S1 Appendix, eTable 1.1 in Appendix 1.

### Study characteristics

The mean age of study participants ranged between 25.3 to 50.56 years old with male predominance estimated as 64.37%. Four studies were done in Europe [12, 25–27], two studies were carried out in India [28, 29] and one study each was conducted in Mexico [30], Iran [31] and Vietnam [32]. In general, five studies reported administration of dexamethasone together with or without antibiotics [12, 27–29, 32], in which dexamethasone was given 15–20 minutes before or with antibiotics in three studies [12, 28, 32] and within 3 hours after antibiotic initiation in one study [27] whereas one study indicated dexamethasone administration after first dose of antibiotic without specifying the timing of initiation [29]. Four studies compared effectiveness of antibiotics used alone and/or combination in treating bacterial meningitis [25, 26, 30, 31]. The details of study characteristics are given in Table 1, S1 Appendix, Appendix 3, eTable 3.1.

### Study quality

In terms of study quality, four studies had moderate risk of bias [25, 26, 28, 30]. Three studies were found to have low risk of bias [12, 27, 32], while the remaining two studies have high risk of bias [29, 31] (S1 Appendix, Appendix 4, eFigure 4.1). The high risk of bias in the two studies were due to no or the lack of information on allocation concealment.

### Meta-analysis

Pairwise meta-analysis for direct comparison results were shown in S1 Appendix, Appendix 7, eFigure 7.1–7.3. No significant difference between any treatments in pairwise comparison

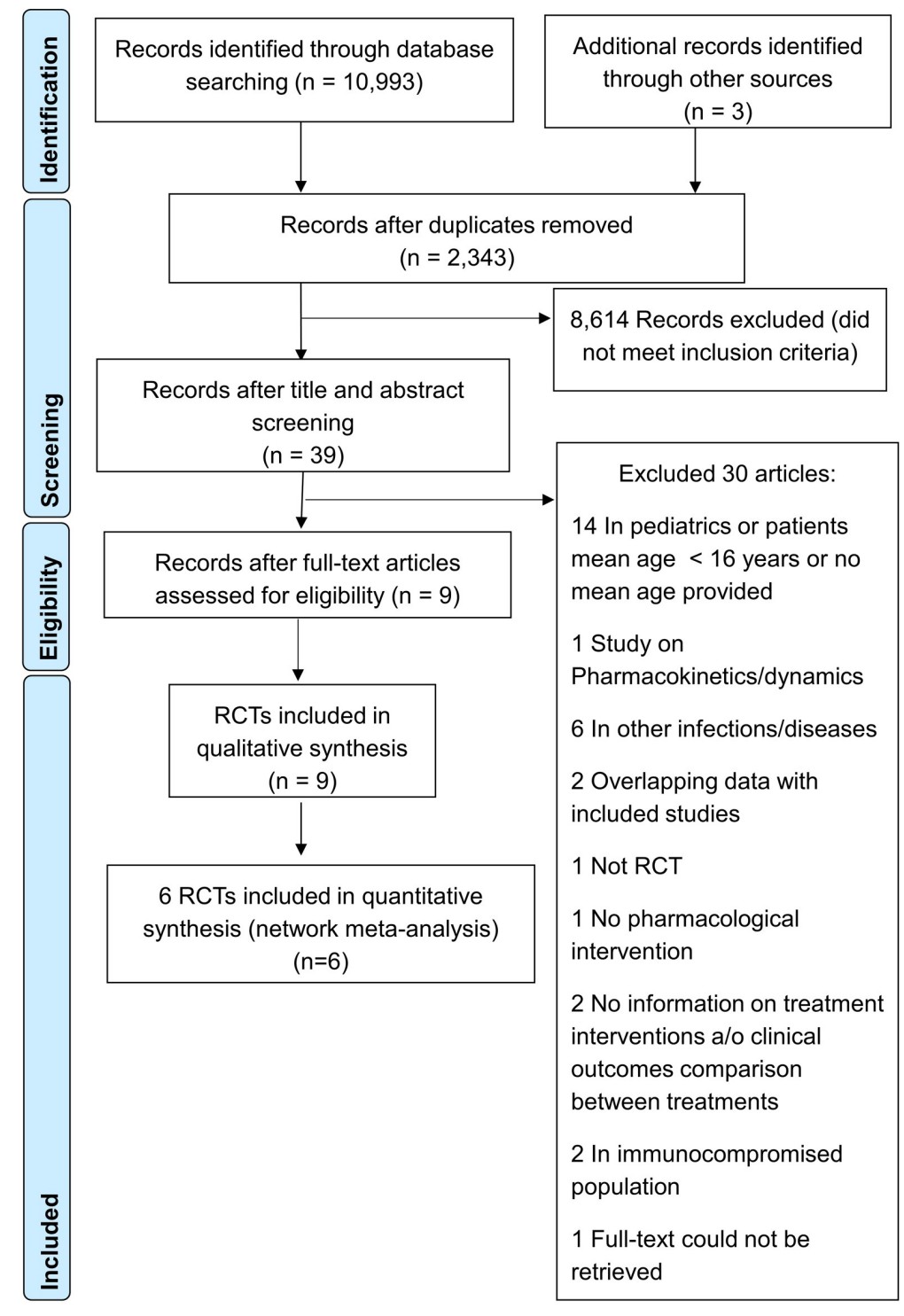

PRISMA, Preferred Reporting Items for Systematic Reviews and Meta-Analyses; RCT, randomized controlled trial

**Fig 1. PRISMA flow chart of study selection process.**

**Table 1. Key characteristics of included studies.**

| Authors (year) | Country | Study design and size (N) (mean ±SD) | Main pathogen (N) | Intervention | Comparator | Primary outcomes |
|---|---|---|---|---|---|---|
| CS+ABT vs. Double | | | | | | |
| Bhaumik, S. (1998)[29] | India | A single center, randomized clinical trial (N = 30) (28±17 vs. 34±19) | *N. meningitis* (n = 6) *S. pneumoniae* (n = 9) *Note*: No of isolates = 15 | Dexamethasone (4 mg q 6 h 4 days then 4 mg tid on 5th day, bid on day 6th-7th) i.v. plus antibiotics ≥ 10 days (CS +PEN+CHLO)(n = 14) *Note*: 13 patients were initially treated with C-pen (20 lac units IV q 4) +Chloramphenical (1 g IV q 6 h) Dexamethasone was started after 1st dose of antibiotics | Antibiotics ≥ 10 days (PEN +CHLO) (n = 16) *Note*: 14 patients were initially treated with C-pen (20 lac units IV q 4) +Chloramphenical (1 g IV q 6 h) | Death 1 (7.1) vs. 3 (18.8), p = 0.60 Neurological sequelae: 3 (21.4) vs. 2 (12.5), p = 0.64 Audiological sequelae*: 4 (28.5) vs. 3 (18.75), p = 1.00 *Note*: * Assessed by brainstem auditory evoked responses a/o pure tone auditory |
| CS+ABT vs. Mono | | | | | | |
| de Gans, J* (2002)[12] | Europe (The Netherlands, Belgium, Germany, Denmark, Austria) | A randomized, double-blind, multicenter trial (N = 301) (44±18 vs 46±20) | *S. pneumoniae* (n = 108, 36%) *Neisseria meningitis* (n = 97, 33% Others (n = 29, 9.7%) Negative CSF culture (n = 65, 21.3%) *Note*: CSF culture was performed in 299 patients (155 vs. 144) | Dexamethasone sodium phosphate 10 mg q 6 h iv, 4 dys 15–20 mins before or with antibiotics (CS+PEN) (n = 157) *Note*: Most patients initially received amoxicillin 2 g iv q 4 h 7–10 dys 77% patients received amoxicillin and penicillin, 8% 3rd generation cephalosporin, 8% amoxicillin or penicillin + cephalosporin | Placebo+antibiotic (PEN) (n = 144) *Note*: Most patients initially received amoxicillin 2 g iv q 4 h 7–10 dys | RR of death, 95%CI: 0.48 (0.24–0.96), p = 0.04 RR of an unfavorable outcome in intervention to comparator, 95%CI: 0.59 (0.37–0.394), p = 0.03 Focal neurologic abnormalities, 18/143 (13) vs. 24/119 (20); 95% CI: 0.62 (0.36–1.09), p = 0.13 Hearing loss 13/ 143 (9) vs. 14/119 (12); 95%CI: 0.77 (0.38–1.58), p = 0.54 |
| Gijwani, D (2002)[28] | India | A prospective placebo controlled, randomized double-blind study (N = 40) (28.25± 16.75 vs 32.25± 1.64) | *Pneumonococci* (n = 8, 40%) *Staphylococci* (n = 6, 30%) *Streptococci* (n = 4, 20%) *H influenzae Streptococci* (n = 4, 20%) *Meningococci* (n = 2, 10%) E. Coli *Meningococci Streptococci* (n = 2, 10%) | Dexamethasone (0.6 mg/ kg/dy q 6 h first 4 days) +ceftriaxone 100 mg/kg/dy 14 days (CS+CEP) (n = 20) *Note*: Dexamethasone was given at least 15 mins before ceftriaxone | Placebo+ ceftriaxone 100 mg/kg/dy 14 days (CEP) (n = 20) | Mortality 2/18 (11.12) vs. 4/16 (25) Neurological sequelae at 90th day: 2/18 (11.12) vs. 4/16 (25) Hearing loss at 90th day: 7/18 (38.89) vs. 9/16 (56.25) |
| Nguyen, TH (2007)[32] | Vietnam | A randomized, double-blind, placebo-controlled trial (N = 435) Median age: 42 (15–89) vs. 41 (15–91) | *S. suis* (n = 116, 52.3%) *S. pneumoniae* (n = 55, 24.8%) *Streptococcus species* (n = 18, 8.1%) *S. Aureus* (n = 9, 4.1%) *N. meningitidis* (n = 19, 8.6%) *H. influenzae* (n = 7, 3.2%) *Klebsiella species* (n = 10, 4.5%) *E. coli* (n = 9, 4.1%) Other -ve bacteria (n = 4, 1.8%) | Dexamethasone 0.4 mg/kg q 12 h, 4 dys 15 mins before antibiotics + ceftriaxone 2 g iv, q 12 h, 10–14 dys (CS+CEP) (n = 217, 143*) *Note*: Antibiotic treatment could be altered based on physician's discretion *Definite bacterial meningitis: If bacteria were detected in CSF or blood culture at the time of discharge or death | Placebo+ ceftriaxone 2 g iv, q 12 h, 10–14 dys (CEP) (n = 218, 157*) *Note*: Antibiotic treatment could be altered based on physician's discretion *Definite bacterial meningitis: If bacteria were detected in CSF or blood culture at the time of discharge or death | Death 1 month after randomization:18/217 (8.29) vs. 26/218 (11.93) All patients:—RR of death at 1 months = 0.79 (0.45–1.39),ns—RR of death or disability at 6 months = 0.74 (0.47–1.17),ns Definite BM:—RR of death at 1 month = 0.43 (0.20–0.94), p = 0.03—RR of death or disability at 6 months = 0.56 (0.32–0.98), p = 0.03 |

(*Continued*)

**Table 1.** (Continued)

| Authors (year) | Country | Study design and size (N) (mean ±SD) | Main pathogen (N) | Intervention | Comparator | Primary outcomes |
|---|---|---|---|---|---|---|
| Thomas, R (1999)[27] | France and Switzerland | Multicenter, double-blind, randomized trial (N = 60) (40±19 vs 50±19) | *S. pneumoniae* (n = 31, 52%) *N. menigitidis* (n = 18, 30%) Unknown (n = 8, 13.3%) Others (n = 3, 5%) | Aminopenicillin + dexamethasone 10 mg qid 3 days (within 3 hours after aminopenicillin therapy initiation) (CS +PEN) (n = 31) *Note*: the first dose of dexamethasone was given within 3 hours after initiation of antibiotics | Aminopenicillin+placebo qid 3 days (within 3 hours after aminopenicillin therapy initiation) (PEN) (n = 29) | The rate of patients cured without neurological sequelae at day 30: 23 (74) vs. 15 (52), p. 0.0711 Mild neurological sequelae at day 30: 2 (6.45%) vs. 4 (13.79%) Severe neurological sequelae: 3 (9.68%) vs. 5 (17.24%) |
| Mono vs. Double | | | | | | |
| Zavala, I (1988)[30] | Mexico | An open, randomized comparative study (N = 26) 28.6 (14–51) vs. 25.3 (16–52). | *S. pnueumoniae* (n = 13) *S. epidermidis* (n = 3) *H. influenza* (n = 2) *E. coli* (n = 6) *S. typhi* (n = 1) | Ceftriaxone i. v. 4 g OD (CEP) (n = 13) *Note*: Dose decreased to 2 g when CSF became sterile | Ampicillin + Chloramphenicol i.v. (PEN+CHLO) (n = 13) *Note*: Ampicillin dose 200–400 mg/kg/dy, chloramphenicol dose 2–3 g/dy in 4 divided doses | The overall clinical and bacteriological cure rate (CSF sterile after 10 days): 100% (13/13) vs. 92% (12/13)* *Note*: *1 patient withdrew due to treatment failure |
| Others | | | | | | |
| Elyasi, S (2015)[37] | Iran | A randomized, open-labeled study (N = 44) (50.56± 18.22 vs 46.13± 17.11) | *S. pneumoniae* (n = 25, 56.82%) *MRSA* (n = 2, 4.54%) *S. epidermidis* (n = 1, 2.27%) *E. faecalis* (n = 1, 2.27%) *Note*: From 29/44 (65.9%) who had positive CSF culture | Vancomycin 15 mg/kg q 8 h (high dose)+Ceftriaxone 2g q 12h (n = 22) *Note*: All patients received 1st dose of antibiotics within 1 h of hospital admission. Most common regimen: Vancomycin plus ceftriaxone 2 g q 12 h (20 vs. 20) Target trough level = 15–20 mg/ml for ABM | Vancomycin 15 mg/kg q 12 h (conventional-dose) + Ceftriaxone 2g q 12 h (n = 22) *Note*: All patients received 1st dose of antibiotics within 1 h of hospital admission Target trough level = 15–20 mg/ml for ABM | GCS at 10th dy: 11 vs. 13, p = 0.02 CrCL at 10th day 102.14±44.24 vs 98.99 ±13.87 ml/min, p = 0.65 Time to normal WBC 3.22±3.11 vs 6.00±2.45, p = 0.03 Time to afebrile 3.35±1.23 vs 6.11±2.00, p = 0.02 Duration of hospitalization day: 12.85 ±5.47 vs 10.10±2.45, p = 0.04 |
| Narciso, P (1983)[38] | Italy | A single-center, randomized controlled trial (N = 10) (48 vs. 44) | *N. meningitidis* (n = 3) *E. coli* (n = 1) *D. pneumoniae* (n = 3) *Note*: Negative culture in 3 patients | Ceftriaxone i.v. q 12 h 100–80 mg/kg/day in 4 cases and 45 mg/kg in 1 case (CEP) (n = 5) | Ampicillin (110 mg/kg) divided in 3 slow iv q 8 hr (PEN) (n = 5) | All patient recovered completely |
| Schmutzhard, E (1995)[39] | Hungary, the Czech Republic, Portugal, France, Spain, Austria (from 15 centres) | Two prospective randomized controlled studies (N = 56) Median 46 (17–76) vs. 31 (13–71) | *H. influenzae* 7.1% (n = 1) *N. menigitidis* 39.3% (n = 18) *S. pneumoniae* 78.6% (n = 31) Others 3.57% (n = 2) | Meropenem 40mg/kg q 8 h, up to a maximum of 6g/dy, 7–14 dys ¶ (MPN) (n = 28) *Note*: Dexamethasone was administered to 39 patients (meropenem 19, cephalosporins 20) | Cephalosporins, 7–14 dys¶ (n = 28): Cefotaxime (n = 17) Ceftriaxone (CEP) (n = 11) *Note*: Dexamethasone was administered to 39 patients (meropenem 19, cephalosporins 20) | Clinical cured: Meropenem vs. Cephalosporins: 23/23 (100%) vs. 17/22 (77%) Neurological sequelae: Meropenem 3 (10.7%) vs. Cephalosporins 4 (14.3%); 4 receiving cefotaxime Hearing impairment: 11 (39.3%) vs. 9 (32%) Death: Meropenem 3 vs. 1 Cephalosporins |

**Abbreviations:** Mono: Mono antibiotic therapy; Double: Dual antibiotic therapy; CS: Corticosteroids; ABT: Antibiotic(s); BM: Bacterial meningitis; C-pen: crystalline penicillin; CSF: Cerebrospinal fluid; FU: follow-up; GCS: GCS: Glasgow Coma Scale score; i.m.: intramuscular; i.v.: intravenous; OD: once daily; SAPS 1: the Simplified Acute Physiologic Score; NR: Not reported; ALT: Alanine transaminase; AST: Aspartate transaminase

were noted. However, there was non-significant favorable trend towards corticosteroids in combination with antibiotic vs. mono antibiotic therapy for all-cause mortality and any hearing loss.

## Network meta-analysis

The contribution plots (S1 Appendix, Appendix 7, eFigure 7.5) of each comparison from individual studies to the overall estimates were presented based on the weighted percentage. For all-cause mortality, corticosteroids in combination with antibiotic vs. mono antibiotic therapy was ranked the most effective, followed by corticosteroids in combination with antibiotic and finally dual antibiotic therapy. Similar results were found for neurological sequelae and any hearing loss with slightly higher contribution percentage for corticosteroids in combination with antibiotic vs. dual antibiotic therapy (42.8% and 44.8%) (S2 data, eFigure 7.5 (b) (c)).

## Network consistency

The geometry network maps of all treatment comparisons for each primary outcomes were presented in Fig 2. According to the global inconsistency test, there was no evidence of inconsistency identified in any loop comparison (S1 Appendix, Appendix 6, eTable 6.1–6.4). Corticosteroids in combination with antibiotic therapy and mono antibiotic therapy were the most frequent regimens investigated (5 of 6 studies) which can be observed from the node size that corresponds to the number of studies examining the intervention (Fig 2).

## Treatment outcomes

**All-cause mortality.** The network meta-analysis of 6 studies in 892 patients examining three treatment options identified no statistically significant superiority in any comparisons (Fig 3A) (S1 Appendix, Appendix 8, eFigure 8.1). The non-significant numerical lower risk of mortality was shown for corticosteroids in combination with antibiotic therapy versus mono

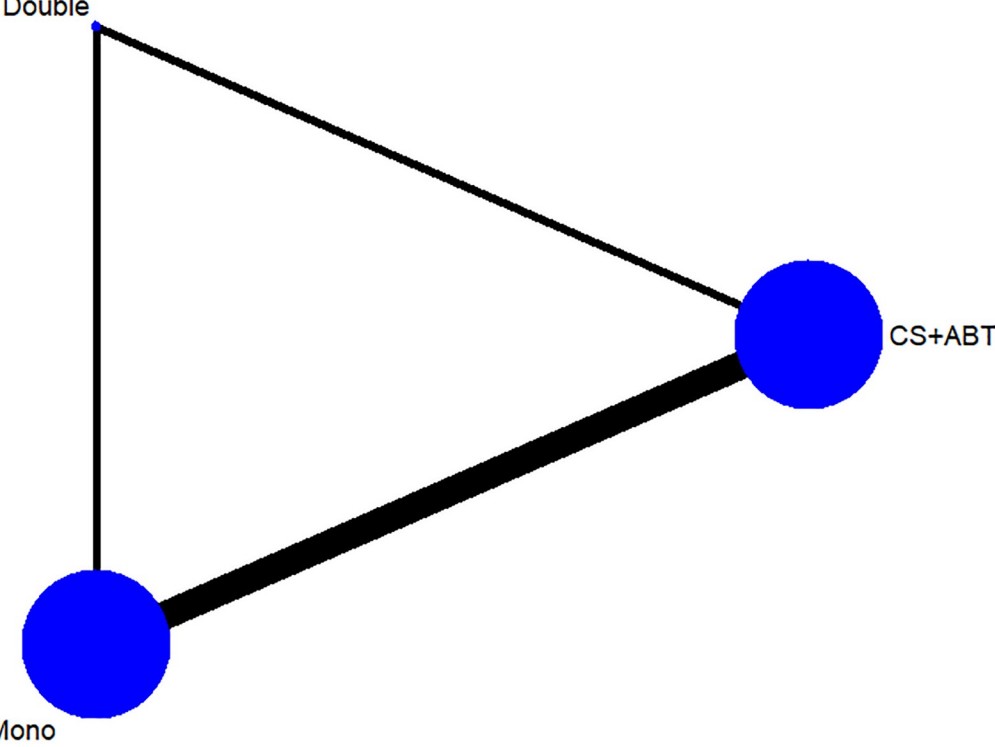

**Fig 2. Network geometry for 3 treatment category options (corticosteroids in combination with antibiotic, dual antibiotic therapy, mono antibiotic therapy).**

antibiotic therapy (RR 0.65; 95%CI, 0.42 to 1.02). Surface under the cumulative ranking area (SUCRA) suggest that corticosteroids in combination with antibiotic, followed by mono antibiotic therapy and finally dual antibiotic therapy (S1 Appendix eFigure 9.1).

There were five studies exploring effectiveness of corticosteroids (i.e. dexamethasone) into treatment regime [12, 27–29, 32]; generally, higher death number was recorded in group treated with antibiotics alone compared to those received combination of dexamethasone and antibiotics. For instance, Gijwani et al. recorded 4 deaths in ceftriaxone group (25%) while only 2 patients died receiving both dexamethasone and ceftriaxone (10%; $p < 0.05$) [28]. Similar results were reported by de Gans et al., whereby 21 patients receiving antibiotics died, while only 11 deaths were observed in patients receiving both dexamethasone and antibiotics [12] (S1 Appendix, Appendix 3, eTable 3.1).

**Neurological sequelae.** The NMA results involving 833 patients found no significant differences observed for any treatment comparison (adjunctive corticosteroids in combination with antibiotic therapy vs. mono antibiotic therapy; RR 0.75; 95%CI 0.47 to 1.18 and dual antibiotic therapy vs. mono antibiotic therapy; RR 0.50; 95%CI 0.10 to 2.47, 6 RCTs, moderate certainty of evidence; dual antibiotic therapy vs. adjunctive corticosteroids in combination with antibiotic therapy; RR 0.67; 95%CI 0.14 to 3.17, 6 RCTs, low certainty of evidence) (Table 2) (Fig 3B) (S1 Appendix, Appendix 8, eFigure 8.2). According to the SUCRA analysis for efficacy ranking, dual antibiotic therapy ranked first, followed by corticosteroids plus antibiotic regimen, and mono antibiotic therapy (S1 Appendix, Appendix 9, eFigure 9.2).

Neurological sequelae were observed in 6 studies [12, 26–29, 32]. The only study (Schmutzhard, E, et. al. 1995) comparing effectiveness of meropenem and cephalosporin treatment reported neurological sequelae in 3 (10.7%) and 4 (14.3%) bacterial meningitis patients [26]. Additionally, five of the studies evaluated effects of adjunctive dexamethasone with antibiotics in patients with bacterial meningitis [12, 27–29, 32]. In the earlier study by Bhaumik and Behari (1998), 3 (21.4%) patients receiving both dexamethasone and antibiotics, and 2 (12.5%) patients received antibiotics alone exhibited neurological sequelae (RR 0.58 (0.11–3.00); p = 0.64). However, dexamethasone was given after antibiotics with no mention about specific timing [29]. Thomas et al. (1999) reported that lower number of patients treated with amoxicillin and dexamethasone within first 3 hours after antibiotic administration showed mild (n = 2, 6.45%) or severe (n = 3, 9.68%) neurological sequelae as compared to control group (mild: n = 4, 13.79%; severe: n = 5, 17.24%) [27]. However, the rate of cured patients without neurological sequelae between the dexamethasone and placebo groups were not statistically significant (p = 0.0711) [27] (S1 Appendix, eTable 3.1, Appendix 3).

On the other hand, de Gans et al. (2002) reported focal neurologic abnormalities in 18 (12.6%) and 24 (20.2%) patients treated with penicillin plus dexamethasone and control group, respectively [12]. Nonetheless, the difference was not significant between the two groups (RR 0.62 (0.36–1.09), p = 0.13). Another study by Gijwani et al. revealed a total of six patients experienced neurological sequelae; lower number of patients receiving dexamethasone treatment in conjunction with ceftriaxone (n = 2, 11.12%) showed neurological sequelae during discharge as compared to control (n = 4, 25%) [28] (S1 Appendix, Appendix 3, eTable 3.1). Similarly a study by Nguyen et al. (2007) reported no beneficial effect observed in patients receiving dexamethasone vs. antibiotic alone observed [32].

**Hearing loss.** According to the network estimated ratio of all treatments for any hearing loss, corticosteroids in combination with antibiotic therapy demonstrated significant lower risk of any hearing loss versus mono antibiotic therapy (RR 0.64; 95%CI, 0.45 to 0.91) (Table 2) (Fig 3C) (S1 Appendix, Appendix 8, eFigure 8.3). There were no significant differences observed for other treatment comparisons although numerical lower risk of hearing loss

**A** Network estimated rate ratios (95% confidence intervals) for all-cause mortality.

| CS+ABT | | |
|---|---|---|
| 0.43 (0.07,2.86) | Double | |
| 0.65 (0.42,1.02) | 1.50 (0.22,10.21) | Mono |

**B** Network estimated rate ratios (95% confidence intervals) for neurologic sequelae.

| CS+ABT | | |
|---|---|---|
| 1.50 (0.32,7.13) | Double | |
| 0.75 (0.47,1.18) | 0.50 (0.10,2.47) | Mono |

**C** Network estimated rate ratios (95% confidence intervals) for any hearing loss.

| CS+ABT | | |
|---|---|---|
| 1.39 (0.40,4.83) | Double | |
| **0.64 (0.45,0.91)** | 0.46 (0.13,1.66) | Mono |

**Fig 3.** Network estimated risk ratio (95% CIs) of treatment comparisons for (a) all-cause mortality, (b) neurologic sequelae, and (c) any hearing loss.

**Table 2. GRADE quality assessment of treatment comparisons for primary outcomes in adult bacterial meningitis.**

| Treatment Comparison | Direct evidences | | Network meta-analysis | |
|---|---|---|---|---|
| | Risk ratio (95% CI) | Quality of evidences | Risk ratio (95% CI) | Quality of evidences |
| **All-cause mortality** | | | | |
| **CS+ABT vs. Double** | 0.45 (0.05 to 3.9) | ⊕⊕○○ LOW | 2.62 (0.30 to 22.77) | ⊕○○○ VERY LOW |
| **CS+ABT vs. Mono** | 0.95 (0.55 to 1.62) | ⊕⊕⊕○ MODERATE | 0.65 (0.42 to 1.03) | ⊕⊕⊕○ MODERATE |
| **Double vs. Mono** | - | ⊕⊕⊕○ MODERATE | 0.58 (0.01 to 49.59) | ⊕⊕○○ LOW |
| **Neurological sequelae** | | | | |
| **CS+ABT vs. Double** | 1.93 (0.37 to 10.01) | ⊕○○○ VERY LOW | 0.67 (0.14 to 3.17) | ⊕⊕○○ LOW |
| **CS+ABT vs. Mono** | 0.95 (0.55 to 1.62) | ⊕⊕⊕○ MODERATE | 0.75 (0.47 to 1.18) | ⊕⊕⊕○ MODERATE |
| **Double vs. Mono** | - | ⊕⊕⊕○ MODERATE | 0.50 (0.10 to 2.47) | ⊕⊕⊕○ MODERATE |
| **Any hearing loss** | | | | |
| **CS+ABT vs. Double** | 1.81 (0.48 to 6.83) | ⊕○○○ VERY LOW | 0.72 (0.21 to 2.49) | ⊕⊕○○ LOW |
| **CS+ABT vs. Mono** | 0.80 (0.55 to 1.14) | ⊕⊕⊕○ MODERATE | 0.64 (0.55 to 1.62) | ⊕⊕⊕○ MODERATE |
| **Double vs. Mono** | - | ⊕⊕⊕○ MODERATE | 0.72 (0.21 to 2.49) | ⊕⊕⊕○ MODERATE |

Abbreviations: Mono, Mono antibiotic therapy; Double, Dual antibiotic therapy; CS, Corticosteroids; ABT, Antibiotic(s); CI, Confidence interval; RR, Risk ratio

in dual antibiotic therapy vs. mono antibiotic therapy was observed but it was not statistically significant (RR 0.46; 95%CI, 0.13 to 1.66).

With regards to the SUCRA analysis for efficacy ranking, dual antibiotic therapy ranked first, followed by corticosteroids plus antibiotic treatment, and mono antibiotic therapy (eFigure 9.3).

Hearing loss event was reported in six studies. Out of six studies, four studies evaluated occurrence of hearing loss associated with usage of dexamethasone in conjunction with antibiotic [12, 28, 29, 32]. A total of 34 patients reported development of hearing loss after receiving penicillin and/or chloramphenicol with or without dexamethasone and without dexamethasone (n = 17 each group) [12, 29]. Lower occurrence of hearing loss was reported in patients receiving dexamethasone treatment with antibiotic experienced hearing loss (n = 28), which is much lesser than those receiving antibiotic alone (n = 46) [28, 32]. For the study comparing meropenem and cephalosporin as treatment of meningitis patients, 11 patients (39.3%) showed hearing loss when treated with meropenem, slightly more than those treated with cephalosporin (n = 9, 32%) [26].

**Adverse events.** Adverse events were noted in 6 studies [12, 26–28, 30, 32]. Gastrointestinal bleeding occurred in 30 patients, sixteen patients were those receiving corticosteroids in combination with antibiotic therapy [12, 28, 32] whereas there were 13 patients treated with mono antibiotic therapy experiencing the event [12, 27, 28, 32]. The occurrence of psychiatric symptoms was noted in 2 patients receiving adjunctive dexamethasone and ceftriaxone [28]. Generally, the number of adverse events were slightly more frequent among those treated with antibiotic plus adjunct corticosteroids compared to antibiotics alone. However, in the study comparing the efficacy between C-penicillin in combination with chloramphenicol with or without dexamethasone by Bhaumik and Behari (1998), there was not any steroid side effect reported [29]. A summary of major adverse events is described in S1 Appendix, Appendix 3, eTable 3.1.

Herpes simplex infection was found among 74 patients in 3 studies without significant difference between different treatment options [12, 27, 32]. Six patients treated with penicillin antibiotics and adjunctive dexamethasone were affected with Herpes labialis [12] while there were 4 patients treated with penicillin antibiotics without steroids experiencing the same event [12]. Similarly, in study by Nguyen, TH et. al. 2007, the number of patients infected with Herpes simplex virus between cephalosporins with and without adjunctive dexamethasone arms were 33 and 30 respectively [32]. One episode each of inflammation at the injection site and rash and neuropathy were noted in meropenem treatment [26]. There were 15 patients affected by fungal infection, 8 patients treated with amoxicillin plus adjunctive dexamethasone, 4 among those treated with amoxicillin [12], and three in meropenem treated patients [26]. Three patients developed skin rashes and diarrhea respectively among those treated with ampicillin or ampicillin plus chloramphenicol [30]. Overall, there was no significant difference between treatments in terms of safety.

**Pathogens.** The detection of pathogens are mainly done through bacterial culture using CSF [12, 25–32]. Out of 667 cases, majority of them identified *Streptococcus pneumoniae* as the main causative agent (n = 263, 39.4%), followed by *Neisseria meningitidis* (n = 153, 22.9%). However, there is one study conducted in Vietnam showed slightly different findings where the highest number of cases were associated with *Streptococcus suis*, contributing 52.3% of total cases in the study (n = 116) [32]. Apart from members of *Streptococcus* and *Neisseria* genera, some studies have identified pathogens such as *Enterococcus faecalis*, *Escherichia coli*, *Klebsiella* sp., *Haemophilus influenza* and *Listeria monocytogenes* [12, 25–28, 30–32].

**Publication bias.** According to the comparison-adjusted funnel plots, there was no sign of asymmetry found in any outcome (S1 Appendix, Appendix 10). According to the trim-and-

fill method examining publication bias in 6 studies for any hearing loss, the result for fixed effect model (S1 Appendix, Appendix 11, Table 11.3) showed a significant summary estimate (p = 0.048) while a non-significant estimate was shown for the random effects model (p = 0.081). As for all-cause mortality and neurological complication, both fixed and random effect models resulted in non-significant summary estimates suggesting no detection of publication bias.

## Certainty of evidence

According to the GRADE assessment, the certainty of evidence was moderate whether corticosteroids in combination with antibiotic was more effective compared to mono antibiotic treatment in all primary outcomes in adult ABM whereas the confidence of evidence was low or very low for corticosteroids in combination with antibiotic therapy vs. dual antibiotic therapy (Table 2). For dual antibiotic therapy compared to mono antibiotic therapy, the risk ratios were not estimable for direct evidence while there was non-significant NMA estimates favouring dual antibiotic therapy vs. mono antibiotic therapy with moderate certainty of evidence for neurological complications and any hearing loss and low certainty of evidence for all-cause mortality. The results were generally consistent for NMA particularly for adjunctive corticosteroids with antibiotic treatment vs. mono antibiotic therapy. There was incoherence between direct and NMA estimates for corticosteroids in combination with antibiotic therapy compared to dual antibiotic treatment in which the certainty of evidence was low or very low (Table 2). The detail of assessment was illustrated in S1 Appendix, Appendix 10, Tables 10.1 and 10.2.

## Discussion

In this systematic review and network meta-analysis, we have combined both direct and indirect evidences to evaluate relative efficacy between adjunctive corticosteroids with antibiotic treatment, dual antibiotic therapy and mono antibiotic treatment in adult with acute bacterial meningitis in terms of all-cause mortality, neurological complications and any hearing loss. The NMA results demonstrated significant lower risk of any hearing loss in antibiotic with adjunctive corticosteroids treatment compared to mono antibiotic therapy whereas there was no significant difference between other treatment comparisons. This might be due to a small number of studies and patients could be included.

As hearing loss occurs at the early stage of meningitis, delayed treatment unlikely provides benefits when there is already permanent damage from suppurative labyrinthitis. Effectiveness of adjunctive corticosteroid probably depends on timing of administration. Majority of patients included in our selected studies were given dexamethasone 15–20 minutes before or together with first dose of antibiotic except in two studies with relatively small number of patients in which dexamethasone was given 3 hours and unspecified timing after antibiotic initiation [27, 29]. In the analyses, there was only one study [12] from high income country whereas the rest were from low and low middle income countries. This is inconsistent with the conclusion from the previous systematic review and meta-analysis that adjunctive corticosteroids was recommended only in high income countries for adult acute bacterial meningitis in terms of hearing loss and neurological complications benefits [13]. Their findings showed a substantially higher mortality rate in studies in low-income countries and adult population which might be partly due to the inclusion of RCTs in HIV patients. The difference findings may be due to difference in comorbidities rather than countries' socioeconomic status.

In the treatment relative ranking, corticosteroids in combination with antibiotic ranked first for all-cause mortality outcome while dual antibiotic therapy ranked first for neurological

complication and any hearing loss. This is due to the lack of studies which had reported the latter outcomes (8 studies for all-cause mortality vs. 6 studies for neurological sequelae and 4 studies for any hearing loss). As such, the evidence from SUCRA ranking is of low certainty and therefore could be unreliable since the magnitudes of difference between effect estimates are not accounted [33]. Under such we suggest that the interpretation of results should be on NMA findings and GRADE analysis (Table 2).

The causative pathogen is an important factor of treatment consideration. A large number of patients with *Streptococcus suis* in the study from Vietnam [32] reflects the high prevalence of *S. Suis* infection in Southeast Asia. In a meta-analysis by McIntyre PB, et. al. (1997), the beneficial effect of adjuvant dexamethasone before or together with antibiotics was observed among pediatric bacterial meningitis with severe hearing loss from *Haemophilus influenza type b* and *pneumococci* [34]. However, the causative pathogens among adult and children ABM are dissimilar. *H. influenzae* and *S. pneumoniae* were the most frequent pathogens found among children with bacterial meningitis [34] whereas *Streptococcus pneumonia*, followed by *Neisseria meningitides* were the most common organisms found among adult bacterial meningitis according to our SR and NMA findings. Anyhow, none of regimens containing adjunctive corticosteroids was associated with harm in our NMA. Therefore, it should be justified to provide adjunctive corticosteroids in community acquired adult bacterial meningitis particularly among those infected with pneumococcal and streptococcal meningitis due to a high likelihood of complications especially hearing loss.

Optimizing treatment in ABM has become a great challenge during the past few years due to the emergence of multidrug-resistant strains [35]. Using mono antibiotic therapy with penicillin or cephalosporins as empirical treatment should not be recommended in bacterial meningitis caused by *S. pneumoniae* in settings with high prevalence of penicillin resistance. New antibiotics including vancomycin could be a viable option in such situations. However, administration with another antibiotic usually a cephalosporin is required due to its poor CSF penetration [36]. In addition, a high dose to achieve a therapeutic trough level between 15 to 20 mg/dl was recommended [35]. The efficacy of conventional (15 mg/kg q 12 h) versus high dose vancomycin (15 mg/kg q 8 h) in combination with ceftriaxone was investigated in an open-labeled RCT [31]. The high dose vancomycin group exhibited significant favorable clinical responses and GCS at 10[th] day without increasing the risk of nephrotoxicity [31]. This suggests that high dose vancomycin plus ceftriaxone regime seems to be a good alternative in settings where there is a high prevalence of penicillin resistance.

To the best of our knowledge this is the first systematic review and network meta-analysis comparing efficacy between different pharmacological treatments focusing on adult bacterial meningitis. Comprehensive searches in nine relevant databases were carried out with rigorous review without time and language restrictions. Exhaustive quality assessment, data extraction and analyses were done with confirmation by at least among two reviewers in order to make the best use of limited evidences. However, some limitations in this analysis could be noted. The wide variety of medication doses and regimens used in different studies make it difficult for treatment comparison. In our study we grouped treatments into adjunctive corticosteroids in combination with antibiotic, dual antibiotic and mono antibiotic treatments under assumption that the combination of treatment should be superior to single antibiotic therapy. In addition, different types of antibiotics were used within the same regimen among studies in which there were some regimens switching. In our case, we defined the treatment between arms based on the majority or more than 75% of patients treated with the specified therapies. The scarcity of RCTs in adult bacterial meningitis resulted in a limited number of studies could be included in the NMA. As a result, subgroup analyses including among high vs. low- and middle-income countries and studies with high, moderate and low risk of bias would not be

possible due to insufficient power. Due to our restrictive study selection criteria in including only adult population with acute bacterial meningitis and excluding those with HIV or immunocompromised conditions, the results may not be applicable in these settings. However, the selection criteria are justified based on our discretion as the study focuses mainly in community acquired meningitis in adult population in which the treatment and causative pathogens are different from viral and nosocomial bacterial meningitis. Therefore, including those patients probably would have caused more heterogeneity and conflicting findings. In addition, the inclusion criteria particularly patients' age are varied between studies. Nonetheless, the overall mean age ranged from 25.3 to 50.56 years and there was no significant heterogeneity identified. A small number of studies with heterogeneity of treatments might be the reason of some non-significant findings. Finally, different primary outcomes and measures were used and there was no event in either of our primary outcomes reported in some studies in which we imputed these outcomes as no event. This should be acceptable based on the authors' justification, otherwise they should have been reported by researchers.

## Conclusion

Adjunctive corticosteroids in combination with antibiotic seems to be more effective than dual or mono antibiotic therapy in reducing the risk of any hearing loss in adult patients with acute bacterial meningitis (ABM). The likely benefit is anticipated if corticosteroids is given before or together with antibiotic treatment. Timely and sufficient treatment are essential to the disease outcomes. In settings with high prevalence of resistant pneumococcal strains (to penicillin and cephalosporins), dual antibiotic therapy (vancomycin in combination with a cephalosporin) should be opted rather than mono antibiotic therapy. More RCTs in adult bacterial meningitis is needed to assure the effect of pharmacological treatments on mortality and morbidity. Future research to provide a better understanding of acute bacterial meningitis mechanism and corticosteroids effects on the disease would be helpful in identifying optimal treatment strategies.

The type of organism, population, and epidemiologic pattern of drug resistance are important factors for treatment consideration in ABM. Microbiological culture and/or Gram's staining and the minimum inhibitory concentration (MIC) test should be mandated for optimizing treatment selection.

## Supporting information

**S1 Checklist. PRISMA NMA checklist.**
(DOCX)

**S1 Appendix.**
(DOCX)

## Acknowledgments

We thank Dr. Sajesh K. Veettil of Department of Pharmacy Practice, International Medical University (IMU), Malaysia for his guidance on the analyses, Assistant Professor Wasan Katip of the Faculty of Pharmacy, Chiang Mai University, Thailand for his clinical advice regarding antimicrobial regimen, Ms. Aniza Haji Ahmad and Mr. Tengku Mohd Suhaimi Raja Abdullah of Monash University Malaysia Library in helping us with some full-text articles retrieval.

## Author Contributions

**Conceptualization:** Ajaree Rayanakorn, Tahir Mehmood Khan, Surasak Saokaew, Shaun Wen Huey Lee.

**Data curation:** Ajaree Rayanakorn, Hooi-Leng Ser.

**Formal analysis:** Ajaree Rayanakorn, Shaun Wen Huey Lee.

**Funding acquisition:** Kok-Gan Chan, Learn-Han Lee.

**Investigation:** Ajaree Rayanakorn, Hooi-Leng Ser, Shaun Wen Huey Lee.

**Methodology:** Ajaree Rayanakorn, Hooi-Leng Ser, Surasak Saokaew, Shaun Wen Huey Lee.

**Supervision:** Bey Hing Goh, Shaun Wen Huey Lee, Learn-Han Lee.

**Visualization:** Ajaree Rayanakorn.

**Writing – original draft:** Ajaree Rayanakorn, Hooi-Leng Ser.

**Writing – review & editing:** Ajaree Rayanakorn, Priyia Pusparajah, Kok-Gan Chan, Bey Hing Goh, Tahir Mehmood Khan, Surasak Saokaew, Shaun Wen Huey Lee, Learn-Han Lee.

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
