## [Decision Letter · Decision Letter 0]

24 Feb 2020

PONE-D-19-30432

Comparative efficacy of different pharmacotherapies for adults with acute bacterial meningitis: a systematic review and GRADE analysis

PLOS ONE

Dear Ms. Rayanakorn,

Thank you for submitting your manuscript to PLOS ONE. After careful consideration, we feel that it has merit but does not fully meet PLOS ONE’s publication criteria as it currently stands. Therefore, we invite you to submit a revised version of the manuscript that addresses the points raised during the review process.

We would appreciate receiving your revised manuscript by Apr 09 2020 11:59PM. To enhance the reproducibility of your results, we recommend that if applicable you deposit your laboratory protocols in protocols.io, where a protocol can be assigned its own identifier (DOI) such that it can be cited independently in the future. For instructions see: http://journals.plos.org/plosone/s/submission-guidelines#loc-laboratory-protocols

We look forward to receiving your revised manuscript.

Kind regards,

Ahmed Negida, MD

Academic Editor

PLOS ONE

Journal Requirements:

2. Please ensure that your latest search was performed in the past 12 months, or justify the search period used in this study.

"This work was financially supported by External Industry Grants from Biotek Abadi Sdn Bhd (vote no. GBA-808138 and GBA-808813) awarded to L-HL, University of Malaya for Research Grant (GA001-2016,

GA002-2016, and PPP Grant no. PG133-2016A) awarded to K-GC."

Reviewers' comments:

Reviewer's Responses to Questions

**Comments to the Author**

1. Is the manuscript technically sound, and do the data support the conclusions?

Reviewer #1: Yes

Reviewer #2: Yes

Reviewer #3: Yes

Reviewer #4: Yes

2. Has the statistical analysis been performed appropriately and rigorously? 

Reviewer #1: Yes

Reviewer #2: Yes

Reviewer #3: Yes

Reviewer #4: Yes

3. Have the authors made all data underlying the findings in their manuscript fully available?

Reviewer #1: Yes

Reviewer #2: Yes

Reviewer #3: Yes

Reviewer #4: Yes

4. Is the manuscript presented in an intelligible fashion and written in standard English?

Reviewer #1: Yes

Reviewer #2: Yes

Reviewer #3: Yes

Reviewer #4: Yes

5. Review Comments to the Author

Reviewer #1: I have no comments to disclose to any author. The manuscript was technically and literally sound to me and I would like to thank the authors as well as the journal members for giving me the chance of reviewing such an article.

Reviewer #2: The study highlights the importance of adjunct treatment of corticosteroid along with appropriate antibiotics in the treatment of bacterial meningitis to prevent the complications. The main age group of that study is the adult unlike most of the studies which focus on children.

I think more tables and figures should be added eg; table for the demographic data for the included studies, the type of organism, and the treatment regimen, however, I see that meta-analysis will be a good add for the guidelines in the management of acute bacterial meningitis.

I also suggest if one of the biostatisticians reviews the statistical part of this meta-analysis and gives his feedback about the figure part.

Reviewer #3: Line 108: Sentence, "comprehensively examine the relative efficacy between corticosteroid" should be "......relative efficacy of corticosteroid"

Line 110: It is no appropriate to say synthesizing results". You can say "Processing or analyzing results..." instead

Line 417-418: Sentence, "To the best of our knowledge .........focusing in..." should be '...focusing on.."

Reviewer #4: The authors performed a systematic review and network meta-analysis to evlauate the benefit of adding corticosteroids to antibiotics. The search and analysis methods are proper (despite the old search date). I have some comments:

* Title: After reading the manuscript, the title should be changed to "Antibiotics alone or in combination with corticosteroids in adults with acute bacterial meningitis. This is because these were the only two modalities investigated in this study. Also, the title should mention this is a network meta-analysis. The short title needs to be shorter.

* Abstract

- Why did you phrase hearing loss as more important than neurological complications? Please explain.

- The search date is two years old. This needs to be updated.

- The results should indicate the presented findings are in comparison with mono or double antibiotics!

- The abstract should present briefly the results of pairwise meta-analysis for the three main outcomes, followed by NMA.

- Also, the effect estimates and 95% for all primary outcomes should be reported!

- Lines 66-68: This outcome is not significant in the results and should be identified here as such. This is benefit of adding numbers!

- In the conclusion, you only mention hearing loss outcome! It is the only significant outcome, but other outcomes should not be ignored, just because they are insignificant.

* Introduction: well-written and provides a rationale that sets the stage for the network meta-analysis. I would just recommend because most readers are not yet familiar with NMA to add some information about it before the objective.

* Methods

- The methods are well-written and is detailed enough to allow replication of the analysis.

- As mentioned above, the search date is too old (two years) and will be more if published. Therefore, the search must be updated!

- PubMed is a search engine that also searches Medline. So, you mentioned you searched Medline and PubMed. So, what search engine you used for Medline?

- Line 130: "Will be" is the way of writing protocols!

- Lines 137-138: Any response from these authors?

- No information on how screening was done are presented!

- Outcomes are well-defined. The same is true for quality assessment and data analysis.

- Have you attempted P score ranking?

* Results

- Lines 231-233: It is not clear which outcome is superior to others!

- The authors should insert the numbers of effect estimate and significance measures to the NMA results section.

- As per my understanding, all pairwise meta-analyses were non-significant, so did all NMAs, except for hearing loss. In terms of SCURA raning, Cs plus ABs ranked first in one outcome, while double antibiotics ranked first in the two other primary outcomes. Although the analysis methods seem proper, it is hard to derive a consistent clinical implication that recommends adding Cs to ABs based on these outcomes. My advice is to repeat the search and try to include the more recent studies to see how the final analysis would turn out.

* Discussion

- Lines 371-374: The authors keep repeating the fact of lower numerical RR despite non-significance. In my opinion, it is just non-significant.

- Again, interpreting a solid consistent clinical implication based on the above-mentioned results is difficult.

- The discussion should be better structured into shorter, more focused paragraphs.

- Following the limitations, add your recommendations for future practice and clinical research.

6. PLOS authors have the option to publish the peer review history of their article (what does this mean?). If published, this will include your full peer review and any attached files.

Reviewer #1: Yes: MM

Reviewer #2: No

Reviewer #3: No

Reviewer #4: No

---

## [Author Response · Author response to Decision Letter 0]

27 Mar 2020

1. The manuscript meets PLOS ONE's style requirements 

2. An updated search from 9 February to 9 March 2020 was performed, and no new studies met the inclusion criteria. 

3. The financial disclosure has also been updated as follows: 

This work was financially supported by External Industry Grants from Biotek Abadi Sdn Bhd (vote no. GBA-808138 and GBA-808813) awarded to L-HL, University of Malaya for Research Grant (GA001-2016, GA002-2016, and PPP Grant no. PG133-2016A) awarded to K-GC.

---

## [Decision Letter · Decision Letter 1]

27 Apr 2020

Comparative efficacy of antibiotic(s) alone or in combination of corticosteroids in adults with acute bacterial meningitis: a systematic review and network meta-analysis

PONE-D-19-30432R1

Dear Dr. Rayanakorn,

We are pleased to inform you that your manuscript has been judged scientifically suitable for publication and will be formally accepted for publication once it complies with all outstanding technical requirements.

With kind regards,

Ahmed Negida, MD

Academic Editor

PLOS ONE

Additional Editor Comments (optional):

Reviewers' comments:

Reviewer's Responses to Questions

**Comments to the Author**

1. If the authors have adequately addressed your comments raised in a previous round of review and you feel that this manuscript is now acceptable for publication, you may indicate that here to bypass the “Comments to the Author” section, enter your conflict of interest statement in the “Confidential to Editor” section, and submit your "Accept" recommendation.

Reviewer #1: All comments have been addressed

Reviewer #2: All comments have been addressed

Reviewer #3: All comments have been addressed

Reviewer #4: All comments have been addressed

2. Is the manuscript technically sound, and do the data support the conclusions?

Reviewer #1: Yes

Reviewer #2: Yes

Reviewer #3: Yes

Reviewer #4: Yes

3. Has the statistical analysis been performed appropriately and rigorously? 

Reviewer #1: Yes

Reviewer #2: Yes

Reviewer #3: Yes

Reviewer #4: Yes

4. Have the authors made all data underlying the findings in their manuscript fully available?

Reviewer #1: Yes

Reviewer #2: Yes

Reviewer #3: Yes

Reviewer #4: Yes

5. Is the manuscript presented in an intelligible fashion and written in standard English?

Reviewer #1: Yes

Reviewer #2: Yes

Reviewer #3: Yes

Reviewer #4: Yes

6. Review Comments to the Author

Reviewer #1: I have no more comments to the author besides the previous ones in the original manuscript. A second future metanalysis applied to a studies in order to increase the sample size yield should be applied.

Reviewer #2: This manuscript is a network metanalysis which compare the effectiveness of antibiotic alone VS antibiotic and corticosteriod in the treatment of acute meningitis. Previously, I reviewed this manuscript and I asked the authors to do some minor comments and they made the necessary changes for the manuscript. Therefore, I support its acceptance for review.

Reviewer #3: No further comments are required. Author has made the required revisions, so it is worthy to be published

Reviewer #4: The authors have effectively addressed my earlier comments. However, I have two comments about the presentation of results in the abstract.

- Line 65 (Clean version): Clarify that this is the results of "Pairwaise" meta-analysis because the reader may think all analysis have been insignificant.

- Lines 68-71 (Clean version): I feel that the authors keep avoiding mentioning that the results of mortality and neurological sequelae have been statistically insignificant.

- Check if the term "mono antibiotic therapy" is commonly used in the literature and see if a more appropriate term can be used throughout the manuscript.

7. PLOS authors have the option to publish the peer review history of their article (what does this mean?). If published, this will include your full peer review and any attached files.

Reviewer #1: Yes: MM

Reviewer #2: No

Reviewer #3: No

Reviewer #4: No

---

## [Editor Report · Acceptance letter]

20 May 2020

PONE-D-19-30432R1 

Comparative efficacy of antibiotic(s) alone or in combination of corticosteroids in adults with acute bacterial meningitis: a systematic review and network meta-analysis 

Dear Dr. Rayanakorn:

I am pleased to inform you that your manuscript has been deemed suitable for publication in PLOS ONE. Congratulations! Your manuscript is now with our production department. 

With kind regards,

on behalf of

Dr. Ahmed Negida 

Academic Editor

PLOS ONE